# Towards a Comprehensive Benchmark for High-Level Synthesis Targeted to FPGAs

**Yunsheng Bai, Atefeh Sohrabizadeh, Zongyue Qin, Ziniu Hu, Yizhou Sun, Jason Cong**
Department of Computer Science
University of California, Los Angeles
{yba,atefehsz,qinzongyue,bull,yzsun,cong}@cs.ucla.edu

## Abstract

High-level synthesis (HLS) aims to raise the abstraction layer in hardware design, enabling the design of domain-specific accelerators (DSAs) targeted for field-programmable gate arrays (FPGAs) using C/C++ instead of hardware description languages (HDLs). Compiler directives in the form of pragmas play a crucial role in modifying the microarchitecture within the HLS framework. However, the number of possible microarchitectures grows exponentially with the number of pragmas. Moreover, the evaluation of each candidate design using the HLS tool consumes significant time, ranging from minutes to hours, leading to a slow optimization process. To accelerate this process, machine learning models have been used to predict design quality in milliseconds. However, existing open-source datasets for training such models are limited in terms of design complexity and available optimizations. In this paper, we present HLSYN, a new benchmark that addresses these limitations. It contains more complex programs with a wider range of optimization pragmas, making it a comprehensive dataset for training and evaluating design quality prediction models. The HLSYN benchmark consists of 42 unique programs/kernels, each of which has many different pragma configurations, resulting in over 42,000 labeled designs. We conduct an extensive comparison of state-of-the-art baselines to assess their effectiveness in predicting design quality. As an ongoing project, we anticipate expanding the HLSYN benchmark in terms of both quantity and variety of programs to further support the development of this field.

## 1 Introduction

In recent decades, the demand for specialized computing systems tailored to specific applications has significantly increased. This has led to the emergence of domain-specific accelerators (DSAs) being implemented in either application-specific integrated circuits (ASICs) or field-programmable gate arrays (FPGAs). By leveraging the unique characteristics of specific workloads, the designer can design DSAs to enhance performance and energy efficiency. This becomes particularly valuable when general-purpose processors like CPUs and GPUs cannot meet the performance or efficiency requirements of certain applications due to the end of Dennard scaling [8, 14]. For instance, Google has developed its custom-designed DSA in the form of an ASIC named the Tensor Processing Unit (TPU) [21], which is highly optimized for machine learning workloads, offering remarkably faster performance and improved energy efficiency compared to CPUs and GPUs. In addition, FPGAs offer a cost-effective alternative with reconfigurability, making them increasingly appealing for accelerating applications across various domains, including search engines and numerous datacenter applications [25, 9, 19], machine learning inference acceleration [15, 17, 32, 1, 27], and autonomous vehicles [7], among others.

37th Conference on Neural Information Processing Systems (NeurIPS 2023) Track on Datasets and Benchmarks.

Nevertheless, the design of DSAs poses distinct difficulties in contrast to general-purpose hardware like CPUs and GPUs [34, 8]. DSAs are commonly developed using hardware description languages (HDLs) at the register-transfer level (RTL), specifically Verilog and VHDL, which are primarily known to circuit designers. To address this challenge, high-level synthesis (HLS) [11, 10] was introduced and is now supported by most EDA (Electronic Design Automation) and FPGA companies [4, 6, 20, 24, 29]. HLS raises the level of design abstraction to C/C++/OpenCL/SystemC, enabling designers to describe the high-level behavioral representation of their designs rather than the low-level data transition in RTL. This abstraction eliminates the need for explicit clock scheduling specifications in the HLS code. Instead, HLS tools analyze the behavior description to schedule operations across different clock cycles, assign operations to available resources, and establish the required control structure. Finally, the HLS tool automatically generates RTL code based on these analyses. It can take several minutes to hours for the HLS tool to generate this RTL code [34]. The RTL can then be passed through logic synthesis and physical design steps, which can consume several hours, to be implemented on the target FPGA. This HLS tool enhances design productivity, shortens design cycles, and allows designers to rapidly explore various design options without the need for manual RTL code writing.

Despite the increased level of design abstraction offered by high-level synthesis (HLS) tools, they still require a considerable amount of hardware design expertise to utilize synthesis directives in the form of pragmas. These pragmas play a crucial role in specifying various aspects of the design, such as memory organization, caching strategies, memory buffer partitioning, parallelization and pipelining of computations, etc. As demonstrated by Chi et al. [8], although the performance of a DSA with no performance-optimizing pragmas can be $108\times$ slower than a CPU, through proper optimization, it can achieve a remarkable performance improvement, surpassing a CPU by $89\times$. However, the optimization process for architecture-specific enhancements is typically limited to hardware programmers and falls beyond the capabilities of the average software programmers. Consequently, there has been a growing focus on automating this optimization process. While some approaches treat the HLS tool as a black box and develop custom-designed heuristics to search through design candidates, a more recent research paradigm leverages machine learning and deep learning techniques. These approaches either learn the behavior of the HLS tool and construct predictive models or employ data-driven exploration methods to search through the solution space. The goal of automating the optimization process is to democratize customized computing and make it more accessible for the average software programmers, allowing them to utilize the tailored hardware acceleration.

To address the need for automating pragma insertion and parameter tuning in high-level synthesis (HLS), machine learning techniques can be used. This approach aims to achieve optimal quality in terms of latency and resource utilization. However, the lack of open-source datasets in this domain and the limitations of existing datasets, which are constrained in terms of design complexity and available optimizations, restrict their practicality. To bridge this gap, this paper introduces HLSYN, the first

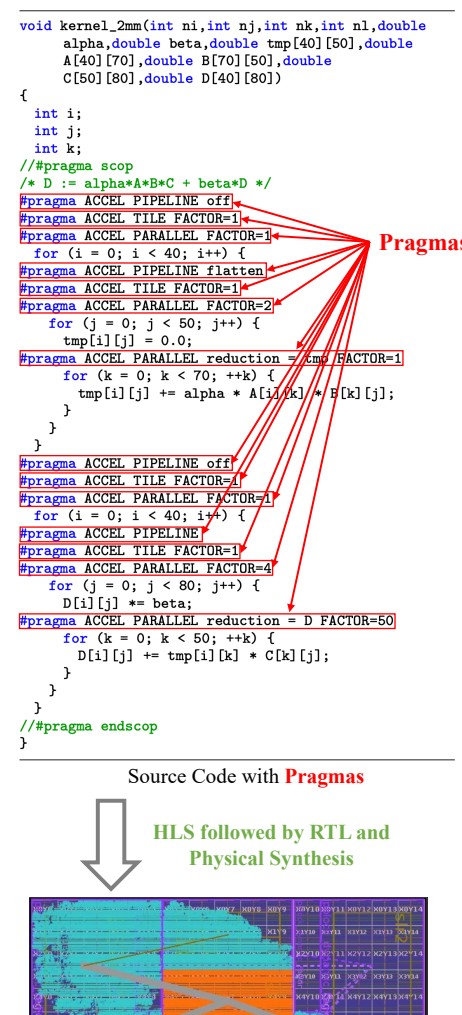

Figure 1: High-level synthesis (HLS) transforms the source code of the kernel 2MM written in C into a lower-level programming language and eventually implements the design on the target FPGA as shown with the chip die photo.

comprehensive benchmark for HLS designs targeted to FPGAs for performance optimization. This benchmark provides more complex programs and a wider range of optimization pragmas, facilitating advanced research and facilitating in-depth exploration of machine learning techniques in the context of HLS. In our study, we define a design as a piece of C/C++ source code (referred to as a kernel) with associated pragmas. Our primary focus is on predicting the quality of a design using supervised learning by running a model trained on a collection of labeled designs with corresponding quality metrics. However, our dataset can be utilized in various training scenarios, including training agents to efficiently explore the solution space.

## 2 Background

The task of HLSYN is to predict the quality of the HLS design specified by a program (kernel) with a specific optimization pragma design. Our target implementation platform is FPGA, although similar techniques can be applied to ASIC accelerator designs as well. The quality of a design is defined as a function of its performance, which is measured by its latency in cycle counts, and its area/resource utilization, such as the usage of digital signal processing blocks (DSP), blocks' RAMs (BRAM), flip-flops (FF), and lookup-tables (LUT), which are the fundamental building blocks for implementing digital logic circuits in FPGA designs.

In this work, we specifically consider the optimization pragmas of the Merlin Compiler, an open-source source-to-source optimization tool used for efficient AMD/Xilinx HLS designs. The Merlin Compiler provides three types of optimization pragmas, namely PIPELINE, PARALLEL, and TILE to define the desired microarchitecture [34].

As illustrated in Figure 1, these pragmas can be applied at the loop level and offer control over the type of pipelining, the parallelization factor, and the amount of data caching. If setting the pragmas properly to non-default parameters for proper parallelizing and pipelining the computation, the resulting accelerator can be $10\times$ or even $100\times$ faster than a single-core CPU. However, without any pragma insertion, the resulting hardware can be $10\times$ slower than a CPU. Figure 2 demonstrates an example where the prediction targets are sensitive to the pragma settings. It is noteworthy that in some other examples, a change in the pragma settings does not lead to any change in the output targets. The machine learning model must learn from existing labeled designs and understand the source code as

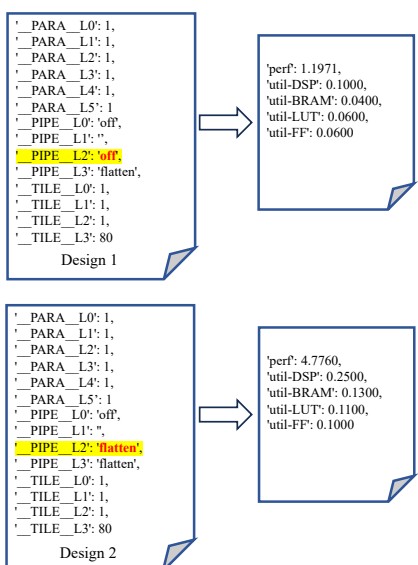

Figure 2: Two example designs selected from the 2MM kernel. In Figure 1, the kernel contains 14 pragmas, and each row has the format "'<pragma name>': <pragma setting>". A small change in one of the pragma parameters leads to changes in the FPGA HLS results, i.e., the five prediction targets for the regression task.

well as the pragmas in order to accurately forecast the outcome of each design when eventually being executed on an FPGA.

Table 1 summarizes the parameter space of these pragmas. For a given program/kernel, any change in the option of any of the pragmas results in a different design with a unique microarchitecture. The "fg" option in pipelining refers to the case where all the inner loops are unrolled (parallelized with separate logic) and each parallel unit is pipelined. The "cg" option, on the other hand, results in coarse-grained processing elements (PEs) that are pipelined together. For example, it can create pipelined load-compute-store units. The PARALLEL and TILE pragma take numeric values that determine the degree of parallelization and loop tiling, respectively.

Table 1: Target pragmas with their options.

| Pragma Name | Parameter Name | Parameter Space | Examples of Pragma Settings |
|---|---|---|---|
| PARALLEL | factor | integer | 4, 8 |
| PIPELINE | mode | "cg", "fg", off | 'flatten' resulting in the "fg" mode |
| TILE | factor | integer | 2, 4 |

# 3 Related Work

In previous research, optimizing HLS designs has been approached in different ways. One category of methods treats the HLS tool as a black box and utilizes problem-independent heuristics or develops dedicated heuristics to explore the solution space and evaluate the quality of results (QoR) directly using the tool [34, 43, 35]. However, this approach is time-consuming as each evaluation takes several minutes to hours. To mitigate this issue, another category of methods aims to create surrogate models for the HLS tool. Some of these methods construct dependency graphs of the program and employ traditional graph analysis techniques to schedule operations and estimate the QoR accordingly [46, 38], while others develop analytical performance and area models to estimate the QoR [45, 47]. Nevertheless, due to the different heuristics employed by HLS tools in the design process, these models may not accurately predict the QoR [34]. Some methods address this limitation by focusing on designs that can exploit pre-defined microarchitecture templates or follow specific computation patterns [33, 37, 12], but this can limit their generality. Alternatively, a data-driven approach utilizing machine learning and deep learning models has been proposed to enhance prediction accuracy [23, 22]. Graph neural networks (GNNs) have gained attention in this context and demonstrated promising results in enhancing prediction accuracy [30, 5, 40, 36].

A fundamental aspect of these approaches is the availability of a large-scale database to effectively train the models. Recent works have focused on gathering such databases [16, 41, 30]. Unfortunately, existing datasets have limitations. The dataset in [41] predominantly consists of synthetic programs that do not utilize any pragmas. DB4HLS [16] targets programs from the MachSuite benchmark[26] but overlooks the inclusion of a key optimization pragma, pipelining. Additionally, DB4HLS views each function in the program as an individual kernel. GNN-DSE [30] targets programs from the Polyhedral benchmark [44], which features more complex kernels for FPGA mapping, in addition to the MachSuite benchmark. This dataset considers a program with all its sub-functions as a kernel, further increasing design complexity. Despite covering a wide range of optimization pragmas for each kernel, the generated dataset is small, with only 9 target kernels and a total of 4,752 data points. To address these limitations, we propose HLSYN, which includes kernels from both the Polyhedral and MachSuite benchmarks. It encompasses a diverse range of optimization pragmas that can pipeline and/or parallelize computation, as well as adjust data caching. Our benchmark is comprised of 42 unique kernels from various domains summarized in Table 2, totaling over 42,000 design points, providing a comprehensive resource for advancing research and facilitating an in-depth exploration of machine learning techniques for HLS.

# 4 The HLSYN Benchmark

In this section, we introduce the datasets in HLSYN[1]. The input data source comes from 42 selected kernels in the MACHSUITE benchmark [26] and the POLYBENCH benchmark suite [44]. Our selected 42 kernels cover a wide range of applications whose descriptions are shown in Table 2.

The benchmark consists of 2 datasets accumulated in the past three years, corresponding to two versions of AMD/Xilinx HLS tools: (1) SDX (V1) [3] and VITIS (V2) [4], with the AMD/Xilinx Alveo U200 as the target FPGA and a working frequency of 250MHz. For each dataset, we select 6 kernels as the held-out testing kernels. They will test the ability of a model to generalize to kernels that it did not see during the training. For the rest of the kernels, we perform a random split with the training, validation, and testing ratio being 70:15:15. Summary statistics of datasets are given in Table 3.

For each dataset kernel in each dataset, we run the two versions of the tools described above to obtain a set of labeled designs. Since the design space size is exponential to the number of pragmas, we rely

---

[1] https://github.com/UCLA-DM/HLSyn

Table 2: There are 42 kernels in total across SDX (V1) and VITIS (V2) spanning multiple domains such as linear algebra on vectors and matrices, data mining, stencil operations, encryption, dynamic programming, etc. **#p** denotes the number of pragmas in the kernel. **# in v1** and **# in v2** denote the number of labeled designs in SDX (V1) and VITIS (V2) respectively.

| Kernel | Source | Description | # pragmas | # in v1 | # in v2 |
|---|---|---|---|---|---|
| 2MM | POLYBENCH | 2 Matrix Multiplications | 14 | 812 | 861 |
| 3MM | POLYBENCH | 3 Matrix Multiplications | 21 | 848 | - |
| ADI | POLYBENCH | Alternating Direction Implicit solver | 13 | 551 | - |
| AES | MACHSUITE | Advanced Encryption Standard | 3 | 45 | 43 |
| ATAX | POLYBENCH | Matrix Transpose and Vector Multiplication | 5 | 884 | 902 |
| ATAX-MEDIUM | POLYBENCH | Matrix Transpose and Vector Multiplication | 5 | 362 | 544 |
| BICG | POLYBENCH | BiCG Sub Kernel of BiCGStab Linear Solver | 5 | 512 | 498 |
| BICG-LARGE | POLYBENCH | BiCG Sub Kernel of BiCGStab Linear Solver | 4 | - | 456 |
| BICG-MEDIUM | POLYBENCH | BiCG Sub Kernel of BiCGStab Linear Solver | 5 | 316 | - |
| CORRELATION | POLYBENCH | Correlation Computation | 17 | 1522 | 699 |
| COVARIANCE | POLYBENCH | Covariance Computation | 13 | - | 356 |
| DOITGEN | POLYBENCH | Multiresolution Analysis | 6 | 179 | 172 |
| DOITGEN-R | POLYBENCH | Multiresolution Analysis | 7 | 595 | 230 |
| FDTD-2D | POLYBENCH | 2-D Finite Different Time Domain Kernel | 16 | 660 | - |
| FDTD-2D-L | POLYBENCH | 2-D Finite Different Time Domain Kernel | 16 | - | 240 |
| GEMM-B | MACHSUITE | Blocked Version of Matrix Multiplication | 9 | 775 | 440 |
| GEMM-N | MACHSUITE | Matrix Transpose and Vector Multiplication | 7 | 749 | 540 |
| GEMM-P | POLYBENCH | Matrix Multiplication | 8 | 1160 | 714 |
| GEMM-P-L | POLYBENCH | Matrix Transpose and Vector Multiplication | 8 | - | 199 |
| GEMVER | POLYBENCH | Vector Multiplication and Matrix Addition | 13 | 924 | 712 |
| GEMVER-M | POLYBENCH | Vector Multiplication and Matrix Addition | 13 | 3365 | - |
| GESUMMV | POLYBENCH | Scalar, Vector and Matrix Multiplication | 4 | 442 | 371 |
| GESUMMV-M | POLYBENCH | Scalar, Vector and Matrix Multiplication | 4 | 304 | - |
| HEAT-3D | POLYBENCH | Heat Equation over 3D Data Domain | 11 | 1664 | - |
| JACOBI-1D | POLYBENCH | 1-D Jacobi Stencil Computation | 5 | 595 | - |
| JACOBI-2D | POLYBENCH | 2-D Jacobi Stencil Computation | 11 | 1862 | - |
| MD | MACHSUITE | n-body Molecular Dynamics | 3 | 12 | - |
| MVT | POLYBENCH | Matrix-Vector Product and Transpose | 8 | 1175 | 1452 |
| MVT-M | MACHSUITE | Matrix-Vector Product and Transpose | 8 | 416 | - |
| NW | MACHSUITE | Dynamic Programming for Sequence Alignment | 6 | 1347 | 615 |
| SEIDEL-2D | POLYBENCH | 2-D Seidel Stencil Computation | 7 | 1314 | - |
| SPMV-CRS | MACHSUITE | Sparse Mat-Vec Mult. w/ Variable-Len. Neighbor | 3 | 114 | 114 |
| SPMV-ELLPACK | MACHSUITE | Sparse Mat-Vec Mult. w/ Fixed-size Neighbor | 3 | 114 | 102 |
| STENCIL | MACHSUITE | A Two-Dimensional Stencil Computation | 7 | 1404 | 1016 |
| STENCIL-3D | MACHSUITE | A Three-Dimensional Stencil Computation | 5 | 239 | 239 |
| SYMM | POLYBENCH | Symmetric Matrix Multiplication | 7 | 153 | 158 |
| SYMM-OPT | POLYBENCH | Symmetric Matrix Multiplication | 8 | - | 281 |
| SYMM-OPT-M | POLYBENCH | Symmetric Matrix Multiplication | 8 | 315 | - |
| SYR2K | POLYBENCH | Symmetric Rank-2k Operations | 8 | 433 | 793 |
| SYRK | POLYBENCH | Symmetric Rank-k Operations | 8 | 660 | 234 |
| TRMM | POLYBENCH | Triangular Matrix Multiplication | 7 | 231 | 968 |
| TRMM-OPT | POLYBENCH | Triangular Matrix Multiplication | 7 | 964 | 281 |

Table 3: Dataset statistics for the input. The meanings of columns are: **#K**: # kernels, **A#K**: average # pragmas per kernel', **A#T**: average # source code tokens per kernel, **A#N**: average # nodes per kernel's graph, **A#E**: average # edges per kernel's graph, **#nt**: # node types, **#pt**: # pragma node types, **nr**: numeric attribute range, **#it**: # instruction type, **#ft**: # flow types, **#bt**: # block types, **#ept**: # edge position types, **#eft**: # edge position type.

| Dataset | #K | A#K | A#T | A#N | A#E | #nt | #pt | nr | #it | #ft | #bt | #ept | #eft |
|---|---|---|---|---|---|---|---|---|---|---|---|---|---|
| SDX (V1) | 37 | 8.0 | 629.9 | 366.7 | 589.6 | 4 | 7 | [0, 494] | 82 | 8 | 56 | 3 | 4 |
| VITIS (V2) | 29 | 7.6 | 629.9 | 334.0 | 534.7 | 4 | 7 | [0, 494] | 74 | 8 | 56 | 3 | 4 |

on heuristics provided by AutoDSE [34] to generate the labels for a subset of all possible designs. The labels come in the form of 5 target values: PERF, DSP, BRAM, LUT, and FF. In addition, according to whether the PERF is greater than a threshold value, we classify a design into two categories: valid and invalid. For the valid designs, we perform the regression task of predicting each one of the 5 target values. Table 4 shows the statistics of the prediction targets, i.e., output of a machine learning model.

Table 4: Dataset statistics for the output. The meanings of columns are **#D for r**: # designs for the regression task, PERF: range of the PERF target, DSP: range of the DSP target, BRAM: range of the BRAM target, LUT: range of the LUT target, FF: range of the FF target, **#D for c**: # designs for the classification task, **T:F**: True class (Valid) vs False class (Invalid) design ratio.

| Dataset | #D for r | PERF | DSP | BRAM | LUT | FF | #D for c | T:F |
|---|---|---|---|---|---|---|---|---|
| SDX (V1) | 9439 | [-6.5, 1.5] | [0.0, 8.4] | [0.0, 3.0] | [0.0, 6.5] | [0.0, 3.2] | 28017 | 9439:18578 |
| VITIS (V2) | 5027 | [-3.6, 6.6] | [0.0, 6.6] | [0.0, 0.7] | [0.0, 5.6] | [0.0, 1.9] | 14273 | 5027:9246 |

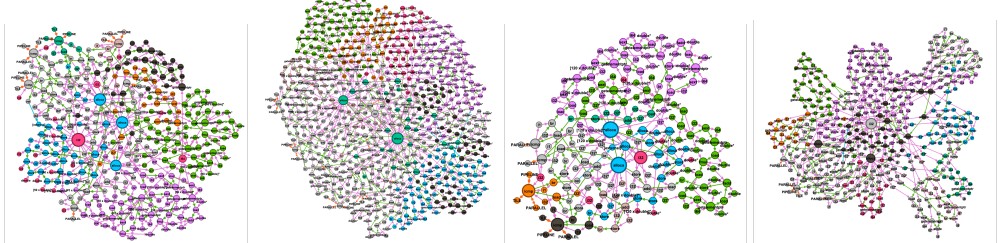

(a) Graph of 2MM consist-ing of 354 nodes and 566 edges.
(b) Graph of CORRELA-TION consisting of 662 nodes and 1071 edges.
(c) Graph of MVT consist-ing of 206 nodes and 328 edges.
(d) Graph of NW consist-ing of 439 nodes and 704 edges.

Figure 3: Visualization of the pragma-augmented PROGRAML graphs of four selected kernels. Node colors indicate the block attribute derived from the assembly code. Edge colors indicate the edge flow.

Since our task aims to predict the validity (classification) and quality (5-target regression) of the designs, we provide both the source code and the graph representation derived from PROGRAML [13]. Specifically, we follow [13] to compile a kernel's source code into assembly code, and then transform the assembly codes into a control data flow graph (CDFG) with call relation, and eventually add pragma nodes to the PROGRAML graph following [30]. The resulting pragma-augmented PROGRAML graphs for four selected kernels are depicted in Figure 3.

It is noteworthy that there are multiple ways to represent the input source programs. As an illustration, we include in this dataset the graph representation used in [30]. Other representations are possible, as such abstract syntax trees. Since we release the source code, it is possible to derive and generate such other representations.

## 5 Experiments on HLSYN

This section provides several baseline experiments with their results to investigate the performance of various methods on the task of design quality prediction.

### 5.1 Baseline Methods

All the baseline methods share the same encoder-decoder architecture and differ only in the encoder. Specifically, each method encodes a design into a $D$-dimensional vector where $D = 512$ by default and uses a MLP-based decoder to transform the design embedding into the targets.

**CODE2VEC [2]** CODE2VEC is a path-based attention model. It first decomposes the code to a collection of paths in its abstract syntax tree and represent them as a bag of distributed vector representations. Then it uses an attention mechanism to compute a learned weighted average of the path vectors as the overall representation of the code.

**CODET5-RAND, CODET5-FROZEN, and CODET5 [39]** These three methods are based on the CODE2VEC method which performs pre-training on a large amount of source code. We utilize the small version released by the authors to encode the design. CODET5 fine-tunes both the encoder and the decoder, CODET5-FROZEN freezes the encoder and only fine-tunes the decoder, while CODET5-RAND fine-tunes both but replaces the encoder parameters/weights with a random initialization to study the effect of pre-training on our tasks.

In order to handle the long source code as a sequence of code tokens, we set the maximum sub-sequence length (i.e., the maximum number of tokens allowed in a sub-sequence) to be 64, and apply a sliding window of size 64 over the source code to obtain multiple sub-sequences as input to the transformer-based encoder. At the beginning of each sub-sequence, a special starting token is inserted and its embedding is taken as the sub-sequence level embedding, and all the sub-sequence embeddings are aggregated into the final $D$-dimensional embedding for the design.

**G-CODEBERT [18] and G-CODEBERT-L** Similar to CODET5, G-CODEBERT is another pre-trained source code encoder, yet with a larger embedding dimension (768 instead of 512), and is thus

Table 5: Regression error (RMSE) on SDx (v1) and VITIS (v2).

| Method | SDx (v1) | | | VITIS (v2) | | |
| --- | --- | --- | --- | --- | --- | --- |
| | Trans | Ind | Ind Adapt | Trans | Ind | Ind Adapt |
| CODE2VEC [2] | 3.2877 | 4.2186 | 3.4156 | 2.9685 | 3.9076 | 2.9957 |
| CODET5-RAND | 1.7100 | 3.2206 | 2.2295 | 1.4939 | 2.9592 | 1.9573 |
| CODET5-FROZEN | 2.6808 | 2.9447 | 2.2317 | 2.1822 | **2.9452** | 1.8447 |
| CODET5 [39] | 0.5515 | 2.8301 | 1.8203 | 0.5270 | 3.4191 | 1.7552 |
| G-CODEBERT [18] | 0.5541 | 2.6639 | 1.5552 | 0.6637 | 3.1158 | **1.5175** |
| G-CODEBERT-L [18] | 0.5651 | 2.7668 | **1.4656** | 0.4910 | 3.2129 | 1.5561 |
| GNN-DSE [30] | 0.8641 | 3.1366 | 1.5180 | 0.5816 | 3.6764 | 1.5546 |
| GNN-DSE-2L | 0.8020 | **2.6587** | 1.6231 | 0.7644 | 3.2825 | 1.5688 |
| [CODET5,GNN-GSE] | **0.4648** | 2.6620 | 1.6327 | 0.4785 | 3.2893 | 1.5274 |
| [G-CODEBERT,GNN-GSE] | 0.5502 | 2.8521 | 1.5525 | **0.3940** | 3.6146 | 1.6017 |

presumably more expressive. G-CODEBERT-L uses a larger maximum sub-sequence length, 128 instead of 64, which would capture a longer dependency between source code tokens, and may yield better results.

**GNN-DSE [30] and GNN-DSE-2L**   In contrast to the previous methods which only receive the source code as input, these methods receive the assembly-level graph (with examples in Figure 3) as input and use TRANSFORMERCONV [28] with a jumping knowledge network [42] to produce the design embeddings. GNN-DSE employs 8 layers whereas GNN-DSE-2L only utilizes 2.

**[CODET5,GNN-GSE] and [G-CODEBERT,GNN-GSE]** These two methods concatenate the design embeddings produced by a source code transformer and a graph neural network-based encoder, i.e., the input embedding into the decoder is $2 \times D$ instead of $D$.

## 5.2   Evaluation Protocol

**Metrics**   There are two tasks for our dataset: regression and classification. The goal of the regression task is to predict the five targets: PERF, DSP, BRAM, LUT, and FF. We use rooted mean square error (RMSE) to evaluate each method. And the goal of the classification task is to predict whether a design is valid or not, i.e. whether the downstream RTL and physical synthesis are likely to complete or not. We use the classification accuracy as the evaluation metric.

In addition, recall that we have two versions of the dataset (SDx (v1) and VITIS (v2)). Each (version,task) combination receives a separate evaluation. For each evaluation, there are two evaluating settings: transductive and inductive testing. Specifically, for each (version, task) combination: (1) We select six kernels as the held-out testing kernels. These kernels are never seen during training and are used for the inductive testing; (2) For the rest of the kernels, we merge all the labeled designs, and randomly split them into training, validation, and transductive testing designs with the 70:15:15 ratio; (3) Using the training designs for 1000 epochs for the regression task, and 200 epochs for the classification task, we train each baseline method. We employ the validation set to determine the best epoch to use for testing; (4) We test the trained model on the transductive testing set. It is called transductive ("**Trans**") since this testing set contains designs from kernels that are seen during training; (5) We test the trained model on the held-out six kernels. Specifically, we (5.1) select the 30% designs from each held-out kernel as the testing designs, (5.2) then repeat the following procedure 5 times. For each kernel, from the rest of the 70% remaining designs, 10 designs are sampled and are utilized to adapt the trained model for 10 epochs. Then, the adapted model is tested on the 30% designs for that held-out kernel. We call such a setting inductive because the model is tested on six kernels that are not visible in the training stage in a zero-shot ("**Ind**") or few-shot learning setting ("**Ind Adapt**").

## 5.3   Results and Analysis

The overall regression and classification results are exhibited in Tables 5 and 6. Tables 7, 8, 9, and 10 reveal the breakdown results over individual held-out kernels.

**Observation 1: There is no consistent winner among the baselines.**   For the regression task, G-CODEBERT and G-CODEBERT-L achieve the lowest error when adapted to the held-out kernels, whereas, for the classification task, GNN-based methods perform better. Such a phenomenon calls for a hybrid model utilizing both the source code and the assembly code graph, e.g., the concatenation models [CODET5,GNN-GSE] and [G-CODEBERT,GNN-GSE]. However, a simple concatenation of the

Table 6: Classification accuracy on SDx (v1) and Vitis (v2).

| Method | SDx (v1) | | | Vitis (v2) | | |
|---|---|---|---|---|---|---|
| | Trans | Ind | Ind Adapt | Trans | Ind | Ind Adapt |
| CODE2VEC [2] | 0.7576 | 0.5662 | 0.6617 | 0.7060 | **0.5444** | 0.6337 |
| CODET5-RAND | 0.8524 | 0.5257 | 0.7015 | 0.7924 | 0.4851 | 0.6291 |
| CODET5-FROZEN | 0.7515 | 0.6098 | 0.7486 | 0.7334 | 0.4161 | 0.6061 |
| CODET5 [39] | 0.9501 | 0.6394 | 0.7447 | 0.9045 | 0.4781 | 0.6734 |
| G-CODEBERT [18] | **0.9536** | 0.6478 | 0.7610 | **0.9233** | 0.5415 | 0.7024 |
| G-CODEBERT-L [18] | 0.9204 | 0.5730 | 0.7701 | 0.8970 | 0.5342 | 0.7180 |
| GNN-DSE [30] | 0.9422 | **0.6529** | **0.7623** | 0.9045 | 0.4781 | 0.6734 |
| GNN-DSE-2L | 0.8912 | 0.6085 | 0.7421 | 0.9126 | 0.5303 | **0.7632** |
| [CODET5,GNN-GSE] | 0.9434 | 0.6141 | 0.7385 | 0.9195 | 0.5053 | 0.7002 |
| [G-CODEBERT,GNN-GSE] | 0.9212 | 0.6174 | 0.7446 | 0.9126 | 0.5001 | 0.7160 |

Table 7: Regression result breakdown on SDx (v1) on individual test kernels.

| Method | DOITGEN-R | FDTD-2D | GEMM-N | JACOBI-2D | STENCIL-3D | TRMM-OPT |
|---|---|---|---|---|---|---|
| CODE2VEC [2] | 2.5711±0.08 | 3.9973±0.19 | 5.6191±0.08 | 2.5710±0.11 | 2.9029±0.04 | 2.8322±0.30 |
| CODET5-RAND | 1.2123±0.02 | 3.3038±0.10 | 3.8933±0.11 | 1.8584±0.12 | 0.9961±0.14 | 2.1133±0.09 |
| CODET5-FROZEN | 1.3197±0.00 | 3.2819±0.06 | 4.3073±0.03 | 1.7756±0.04 | 0.6807±0.03 | 2.0248±0.09 |
| CODET5 [39] | 0.9990±0.27 | 2.9818±0.05 | 3.6910±0.41 | 1.3470±0.12 | 0.3695±0.06 | 1.5336±0.14 |
| G-CODEBERT [18] | 0.8301±0.17 | **2.5938±0.15** | 3.3672±0.45 | 1.1581±0.06 | 0.4830±0.16 | 0.8990±0.19 |
| G-CODEBERT-L [18] | 0.5376±0.08 | 2.8015±0.18 | 3.1310±0.52 | **1.0214±0.12** | 0.3636±0.07 | 0.9384±0.10 |
| GNN-DSE [30] | **0.5278±0.03** | 2.6449±0.10 | 2.8907±0.51 | 1.3855±0.10 | 0.6918±0.23 | 0.9673±0.05 |
| GNN-DSE-2L | 0.9113±0.16 | 2.6677±0.11 | **2.7881±0.28** | 1.5230±0.07 | 0.6984±0.09 | 1.1499±0.05 |
| [CODET5,GNN-GSE] | 0.9637±0.20 | 2.7998±0.30 | 2.9908±0.16 | 1.2991±0.14 | 0.7373±0.14 | 1.0056±0.04 |
| [G-CODEBERT,GNN-GSE] | 1.0409±0.21 | 2.7988±0.17 | 3.2942±0.23 | 1.0845±0.12 | **0.3072±0.06** | **0.7891±0.24** |

Table 8: Regression result breakdown on Vitis (v2) on individual test kernels.

| Method | COVARIANCE | FDTD-2D-L | GEMM-N | GEMM-P-L | SYMM | TRMM-OPT |
|---|---|---|---|---|---|---|
| CODE2VEC [2] | 2.4977±0.09 | 3.1259±0.18 | 4.5526±0.19 | 3.7389±0.03 | 1.4209±0.11 | 2.6379±0.21 |
| CODET5-RAND | 1.6388±0.13 | 2.4587±0.22 | 2.8477±0.15 | 3.0722±0.04 | 0.4969±0.03 | 1.2295±0.09 |
| CODET5-FROZEN | 1.2137±0.02 | 2.4494±0.22 | 2.6576±0.10 | 3.1420±0.02 | 0.3854±0.02 | 1.2199±0.02 |
| CODET5 [39] | 1.3594±0.15 | 2.2958±0.07 | 2.8436±0.37 | 2.7803±0.07 | 0.4092±0.09 | **0.8427±0.14** |
| G-CODEBERT [18] | 1.0652±0.06 | 1.9245±0.09 | 2.3772±0.31 | **2.3913±0.04** | 0.4019±0.08 | 0.9452±0.09 |
| G-CODEBERT-L [18] | **0.9107±0.08** | **1.7740±0.11** | 2.6635±0.33 | 2.5029±0.09 | 0.3384±0.03 | 1.1469±0.11 |
| GNN-DSE [30] | 1.1496±0.04 | 1.8897±0.04 | 2.3157±0.23 | 2.7828±0.18 | 0.4110±0.02 | 0.7790±0.06 |
| GNN-DSE-2L | 1.1457±0.02 | 1.9521±0.09 | **2.0248±0.26** | 2.9691±0.10 | 0.4067±0.04 | 0.9146±0.03 |
| [CODET5,GNN-GSE] | 1.0988±0.07 | 1.8532±0.10 | 2.2430±0.23 | 2.5489±0.04 | **0.3766±0.06** | 1.0440±0.09 |
| [G-CODEBERT,GNN-GSE] | 1.1182±0.07 | 1.9088±0.10 | 2.5352±0.25 | 2.7655±0.08 | 0.3842±0.05 | 0.8986±0.13 |

Table 9: Classification result breakdown on SDx (v1) on individual test kernels.

| Method | DOITGEN-R | FDTD-2D | GEMM-N | JACOBI-2D | STENCIL-3D | TRMM-OPT |
|---|---|---|---|---|---|---|
| CODE2VEC [2] | 0.6449±0.07 | 0.6192±0.01 | 0.4598±0.03 | 0.8122±0.05 | 0.6085±0.03 | 0.8256±0.04 |
| CODET5-RAND | 0.6663±0.05 | 0.5283±0.08 | 0.6366±0.08 | 0.8384±0.03 | 0.5887±0.11 | 0.9509±0.00 |
| CODET5-FROZEN | **0.7472±0.00** | 0.6414±0.00 | 0.6286±0.17 | **0.8889±0.00** | 0.6338±0.00 | **0.9516±0.00** |
| CODET5 [39] | 0.7236±0.04 | 0.6455±0.02 | 0.5804±0.04 | 0.8681±0.01 | 0.7042±0.05 | 0.9467±0.01 |
| G-CODEBERT [18] | 0.6607±0.06 | 0.6455±0.03 | **0.6714±0.03** | 0.8509±0.02 | 0.8000±0.04 | 0.9377±0.02 |
| G-CODEBERT-L [18] | 0.7236±0.08 | 0.6556±0.03 | 0.5741±0.07 | 0.8792±0.01 | **0.8394±0.05** | 0.9488±0.01 |
| GNN-DSE [30] | 0.6506±0.06 | 0.6889±0.02 | 0.6402±0.03 | 0.8652±0.03 | 0.7803±0.03 | 0.9488±0.00 |
| GNN-DSE-2L | 0.6674±0.05 | 0.6778±0.03 | 0.6545±0.02 | 0.8534±0.03 | 0.6648±0.04 | 0.9349±0.03 |
| [CODET5,GNN-GSE] | 0.7022±0.07 | 0.6364±0.02 | 0.6438±0.03 | 0.8419±0.03 | 0.6620±0.04 | 0.9446±0.01 |
| [G-CODEBERT,GNN-GSE] | 0.7124±0.06 | **0.7091±0.04** | 0.6393±0.04 | 0.8616±0.04 | 0.6028±0.09 | 0.9426±0.02 |

Table 10: Classification result breakdown on Vitis (v2) on individual test kernels.

| Method | COVARIANCE | FDTD-2D-L | GEMM-N | GEMM-P-L | SYMM | TRMM-OPT |
|---|---|---|---|---|---|---|
| CODE2VEC [2] | 0.5830±0.05 | 0.5750±0.02 | 0.5099±0.02 | 0.6678±0.05 | 0.7191±0.05 | 0.7476±0.03 |
| CODET5-RAND | 0.6453±0.03 | 0.6444±0.04 | 0.6185±0.03 | 0.6678±0.04 | 0.6936±0.05 | 0.5048±0.25 |
| CODET5-FROZEN | **0.6698±0.00** | 0.4778±0.11 | 0.4951±0.04 | 0.6610±0.00 | 0.5234±0.11 | 0.8095±0.00 |
| CODET5 [39] | **0.6698±0.04** | 0.5750±0.06 | 0.5728±0.03 | 0.6983±0.05 | 0.6766±0.04 | **0.8476±0.03** |
| G-CODEBERT [18] | 0.5585±0.07 | 0.6111±0.07 | 0.5840±0.03 | **0.8373±0.04** | **0.9021±0.05** | 0.7214±0.07 |
| G-CODEBERT-L [18] | 0.5887±0.05 | 0.6583±0.08 | 0.6778±0.03 | 0.8203±0.07 | 0.7702±0.07 | 0.7929±0.02 |
| GNN-DSE [30] | 0.6509±0.01 | 0.7000±0.07 | 0.6840±0.06 | 0.7559±0.05 | 0.8170±0.08 | 0.8381±0.03 |
| GNN-DSE-2L | 0.6396±0.02 | **0.7639±0.10** | **0.7222±0.02** | 0.7763±0.06 | 0.8340±0.06 | 0.8429±0.02 |
| [CODET5,GNN-GSE] | 0.6491±0.06 | 0.6361±0.02 | 0.6494±0.03 | 0.7017±0.04 | 0.8128±0.04 | 0.7524±0.09 |
| [G-CODEBERT,GNN-GSE] | 0.6396±0.03 | 0.6083±0.05 | 0.6691±0.05 | 0.7695±0.03 | 0.8213±0.07 | 0.7881±0.03 |

design embeddings does not consistently yield better performance. Particularly, [CODET5,GNN-GSE] and [G-CODEBERT,GNN-GSE] achieve the lowest regression error under the transductive setting but fall short when adapted to new kernels. Such results imply that future efforts can be made on studying the generalization abilities of machine learning models on our HLSYN benchmark.

**Observation 2: In general, pre-training helps with the performance of source code transformer models.** This can be seen by comparing CODET5-RAND and CODET5, where the former starts training from scratch while the latter loads a pre-trained model released by [39]. This should not come as a surprise, because pre-training has been shown to demonstrate success in natural language processing. Our experimental results verify the effectiveness of pre-training on the HLSYN benchmark. One implication is that one can design better pre-training methods on source code related to electronic design automation, or even design pre-training for GNNs operating on assembly-level graphs.

**Observation 3: More GNN message passing layers does not necessarily improve the performance of GNN models.** In many cases, the 2-layer version, GNN-DSE-2L, performs even better than the 8-layer version, GNN-DSE. This may be attributed to the fact that an attention-based global readout function is used to aggregate node embeddings into a graph-level embedding representing the entire design, and thus each node does not necessarily need to reach far-away nodes in the local message passing stage.

**Observation 4: Generalization to new kernels is difficult, and the performance after adaptation differs across kernels.** For example, G-CODEBERT-L achieves the overall lowest error ("Ind Adapt") on SDX (V1) on the regression task, but Table 7 demonstrates that G-CODEBERT-L does not always yield the lowest error on each of the six held-out kernels. For example, [G-CODEBERT,GNN-GSE] performs the best on STENCIL-3D and TRMM-OPT, but poorly on DOITGEN-R, and thus the average regression error over the six held-out kernels is higher than G-CODEBERT-L. This calls for a more in-depth study of the discrepancy between kernels and a model that is capable of generalizing to a diverse set of kernels. In general, adaptation is necessary, since across all the experiments, without any adaptation ("**Ind**"), directly applying the trained model to new unseen kernels leads to poor regression error and classification accuracy.

## 6    Conclusion and Future Work

This work introduces the task of design quality prediction in the forms of regression and classification tasks and presents the HLSYN benchmark to evaluate state-of-the-art program representation learning methods. Although there is no method that consistently outperforms all the other methods, we notice several trends and identify promising directions toward a more accurate prediction model design. As program representation learning is a continuously growing research domain, we plan to maintain the benchmark to test new methods. For example, our recent work [31] uses hierarchical graphs for program representations to predict design performance. Our HLSYN benchmark is a growing project. We expect to include more kernels and labeled designs running newer versions of HLS tools and establish a leaderboard [2] to encourage participation. In addition, the current benchmark does not consider the design space exploration (DSE) stage, which will be added as the project develops. In fact, the regression task aims to be eventually integrated into the DSE process, which traverses the design space in order to find the optimal pragma setting.

## 7    Acknowledgement

This work was partially supported by NSF 2211557, NSF 1937599, NSF 2119643, NSF 2303037, NASA, SRC JUMP 2.0 Center, Okawa Foundation, Amazon Research, Cisco, Picsart, Snapchat, and CDSC industrial partners (https://cdsc.ucla.edu/partners/). We would also like to thank AMD/Xilinx for HACC equipment donation and Marci Baun for editing the paper.

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
