# OpenReview forum: "Towards a Comprehensive Benchmark for High-Level Synthesis Targeted to FPGAs"
_NeurIPS.cc/2023/Track/Datasets_and_Benchmarks — NeurIPS 2023 Datasets and Benchmarks Poster_

### Official Review · Reviewer_455o · 2023-07-20
**A helpful dataset for FPGA-targeted HLS study**

**Rating:** 7
**Confidence:** 3
**Correctness:** The submitted dataset is constructed …

**Strengths:**

HLSYN encompasses a diverse range of optimization pragmas that can pipeline and/or parallelize computation, as well as adjust data caching. The benchmark is comprised of 42 unique kernels from various domains, totaling over 42,000 design points, providing a comprehensive resource for advancing research and facilitating an in-depth exploration of machine learning techniques for HLS.

The paper & document are detailed and well structured.

**Additional Feedback:**

N/A

**Clarity:**

The paper is well-written and easy to understand even for researchers without HLS background.


**Documentation:**

The documentation is very detailed and organized.

**Ethics:**

No ethical concerns.

**Limitations:**

Currently three optimization pragmas (i.e., PIPELINE, PARALLEL, and TILE) are considered. Can we include more pragma types? Also, there could be more options for each pragma. Of course, we understand that the generation of a larger dataset takes time and we expect the expansion of this dataset in the future.


**Opportunities For Improvement:**

Perhaps not calling it 'first work' in this task in the abstract, considering prior works [16, 40, 30] also release HLS datasets.

Fig 3 may need slightly more explanation.

The authors may explain why the number of pragmas should be fixed in each kernel. Can we choose to turn off PARALLEL/TILE or include more pragma in a few places?

What are the classification/regression result metrics in Tables 7, 8, 9, and 10?

In Section 5.2, the paper mentions "(2) For the rest of the kernels, we merge all the labeled designs, and randomly split them into training, validation, and testing designs with the 70:15:15 ratio;", does it mean the designs from the same kernel can go to training and testing at the same time? This setup is less challenging. But in Section 4, it mentions "For the rest of the kernels, we perform a random split with the training, validation, and testing ratio being 70:15:15". I assume it means all designs from same kernels go to the same place? This conflict is a bit confusing.

The authors may further provide other split methods for training & validation & test & held-out. First, the split may further consider similarities between different kernels. In this way, the ML model can be tested on truly 'unseen' new kernels in held-out. Second, instead of 70:15:15 split, the test set may support cross-validation to enable testing on all samples.

For the quality of a design, besides latency in cycle counts, are WNS or max frequency also worth collecting?


**Relation To Prior Work:**

Related prior works are already well cited in Section 3, especially [16, 40, 30]. The authors are obviously very familiar with this direction.


**Summary And Contributions:**

This work proposes a benchmark named HLSYN for machine learning applications in high-level synthesis (HLS). It consists of 42 unique programs/kernels, resulting in over 42,000 labeled designs.

---

> ### Author Response · Authors · 2023-08-22
> **Thank you very much for your review.**
>
> Our design space marks all the possible pragmas that can be used for each kernel, but each of these pragmas can be turned off (by using factor 1 for parallel and tile pragma and using the keyword `off` with pipeline pragma). This is why we can start with a fixed number of pragmas "placeholders" and then the DSE can decide which pragmas must be enabled.
>
> Regarding your question on WNS or max frequency, we agree that WNS and frequency are important information for assessing the quality of the design. Thank you very much for pointing it out! However, it takes about a day to collect these labels for each point as opposed to minutes to hours for the objectives that we used since we can get them after HLS synthesis, while frequency-related objectives are only accurate after PnR. We plan to gather post-PnR objectives in the future and take advantage of transfer learning techniques to collect a smaller set of samples for them.
>
> The solution space considers all types of pragmas and all valid pragma options (as defined in AutoDSE [TODAES'22]). Please note that we are using the Merlin Compiler, and thus we only need to deal with 3 types of pragmas as the Merlin Compiler applies source-to-source code transformations to generate an equivalent AMD/Xilinx HLS code with respective pragmas that include pipeline, unroll, array_partition, inline, dependence, loop_flatten, etc.

---

> > ### Comment · Reviewer_455o · 2023-08-26
> >
> > Thank you for your response!

---

### Official Review · Reviewer_JA6Z · 2023-07-20
**Well motivated dataset for quality prediction models in HLS**

**Rating:** 7
**Confidence:** 3

**Strengths:**

* Inclusive collection of kernels spanning diverse application domains.
* Evaluation of multiple SOTA design quality prediction methods utilizing the proposed dataset, yielding valuable observations for enhanced model design.
* Provision of quality source code and comprehensive readme for training and testing of the methods.

**Additional Feedback:**

Please refer to the comments in the previous sections.

**Clarity:**

The paper is succinct and exhibits clear writing. Nevertheless, there are multiple instances, such as in Table 4, where insufficient caption spacing hampers readability.

**Correctness:**

The method described in the paper is robust and well-founded; however, it lacks source code detailing the dataset generation process. Nonetheless, the evaluation of design quality prediction is apt and commendable.

**Documentation:**

The dataset's value can be further enhanced by incorporating detailed instructions for dataset generation. While the dataset will be made available, providing an elaborated maintenance plan would be advantageous.

**Ethics:**

There is no ethical concerns.

**Limitations:**

The authors have discussed the limitation of their work in the last section of the main paper.

**Opportunities For Improvement:**

* As stated in the paper, the dataset is subject to continuous development, encompassing additional kernels, pragmas, and designs.
* Detailed instructions and source code can be provided for generating the dataset, ensuring reproducibility and further expansion.

**Relation To Prior Work:**

Related work are clearly discussed.

**Summary And Contributions:**

This paper introduces a dataset tailored for training and evaluating machine learning-based quality prediction models in High-Level Synthesis. The dataset comprises 42,000 labeled FPGA designs, encompassing both performance metrics and resource utilization. These designs originate from 42 distinct programs, compiled using Merlin Compiler and AMD HLS tools. The dataset is employed to assess the performance of ten baseline methods for design quality prediction.

---

> ### Author Response · Authors · 2023-08-22
> **Thank you very much for your response.**
>
> We will do our best to continue working on this project and provide detailed instructions to ensure reproducibility.

---

### Official Review · Reviewer_C7Q3 · 2023-07-24
**Good dataset but the intended use case is limited**

**Rating:** 6
**Confidence:** 4
**Correctness:** Yes.
**Clarity:** It can be improved a little bit more …

**Strengths:**

This work provides valuable insights into how different program feature extraction networks perform in predicting HLS program validity and QoR. For instance,  the study (table 5 and 6) demonstrates that G-CODEBERT models achieve high ind-adapt accuracy in predicting QoR of the HLS programs, while GNN-DSE exhibits high accuracy for classification tasks. It also demonstrates that the generalizability of the networks trained on the dataset is poor, which suggests that a better network or a more comprehensive dataset is needed for training.

**Additional Feedback:**

Please see above.

**Documentation:**

A zenodo link is provided and the work is using Creative Commons Attribution 4.0 International license.

**Ethics:**

No.

**Limitations:**

1. It is unclear how accurate the evaluated ML-based methods are compared to a fast compiler-based analysis to predict the latency and resource utilization (that is present in most of the existing HLS tools, such as AMD Vivado HLS/SDx/VITIS and Intel OpenCL). The paper failed to mention the comparison to the existing non-ML-based analysis in terms of both accuracy and speed.

2. The dataset is composed of programs from existing benchmarks and only targets AMD/Xilinx HLS compiler as the pragmas are critical to performance in these tools. The labels and evaluations are collected on a specific FPGA Alveo U200. It would be more useful to the ML for the compiler community if it targets multiple devices and not only covers various pragmas.

**Opportunities For Improvement:**

1. The diversity of HLS programs in the dataset appears to be limited. As shown in Table 2, more than 15 out of the 42 kernels in the dataset are specifically targeting matrix-matrix or matrix-vector operations. To gain a more comprehensive understanding, it would be beneficial to explore a broader range of workloads and applications in future research.

2. It will be helpful to add the model size in the evaluation to help the readers gain more insights.

3. Showing the trans, ind, and ind adapt metrics can be confusing. The ind adapt should be the only one to present in my opinion. High trans and low ind adapt can just mean the model is overfitted.

4. It will be helpful to explain that there are two versions in the dataset and how the results for sDx and VITIS differ. If the results obtained from both versions showed no significant difference in their ability to derive insights. It might be helpful just to show the results for one version.


**Relation To Prior Work:**

Yes, it mentioned related prior work.

**Summary And Contributions:**

HLSyn presents a dataset comprising 42 High-Level Synthesis (HLS) programs, each compiled with different pragmas. This dataset yields a total of 420,000 labeled designs. The paper provides a thorough comparison of state-of-the-art neural network models used for encoding programs in regression tasks, with the aim of predicting HLS circuit latency and resource utilization on FPGAs.

---

> ### Author Response · Authors · 2023-08-22
> **Thank you very much for your response.**
>
> We will highlight the fact that high trans and low ind adapt can just mean the model is overfitted.
>
> Regarding the difference between the two versions, the results from the different versions of the HLS tool often are different. In fact, among the 22 kernels shared between the two versions, the average latency of the optimal design generated with SDx 2018.3 is 5.54x higher than the ones with Vitis 2020.2.

---

### Official Review · Reviewer_mcz1 · 2023-07-28
**Issues with sampling and creating representative data-set covering the entire search space**

**Rating:** 5
**Confidence:** 5
**Correctness:** The dataset have issues with diversit…
**Clarity:** The paper is clear and written well

**Strengths:**

Authors have combined POLYBENCH and MACHSUITE benchmarks; used pragma configurations existing on the SOTA methods and generated a set of labeled data to be used for training a regressor (instead of using HLS tools to get accurate results)

**Additional Feedback:**

The reviewer suggests the authors to work on issue raised in "Opportunities for improvement" and other minor improvements.

**Documentation:**

The README.md needs improvement. The users need to expert have to troubleshoot issues and run the tool. However, these issues can be easily mitigated.

**Limitations:**

There are minor improvements which authors can easily take care of:

1) Authors claim more complex programs (line 11 and line 92) however they used kernels from Machsuite and polybench which exist in HLS literature. If authors are not including new programs, I suggest dropping the term of more complex programs compared to prior work which is misleading.

2) Line 147, “Nevertheless, due to the different heuristics employed by HLS tools in the design process, these models may not accurately predict the QoR”. Provide citation or detailed explanation on this statement.

3) Line 225: “How are the subsequence embedding aggregated?, Please explain.”

4) Section 5.2 description is *too* much confusing: especially the “Ind-adapt”. When authors say 10 designs out of 70% of designs, do they mean 70% designs out of training data or 70% designs created out of held-out kernels? Based on the realistic assumption of Ind-adapt, I consider 10 designs created out of held-out kernels for adaptation for 10 epochs. But, what does adaptation indicate? Fine-tuning? Please explain.

**Opportunities For Improvement:**

The authors have created a dataset to train the regressor and classifier predicting the QoR of the design. However, the dataset needs to have representative samples out of the massive search space. For e.g. in kernel 2mm, there are 14 locations where pragmas are inserted. Now, assuming each pragma have three possible choices (although PARALLEL and TILE can have more), the search space is $3^{14}$ which is $4,782,969$. Now, the authors have only sampled around 900 out of this massive search space and they relied on heuristics of AutoDSE which uses constrained coordinate optimization leading to samples of good QoRs. Although, as per authors in [1], the sampling strategy covers wide variation in labels, however, AutoDSE misses sampling the designs which can have wide variation in pragmas resulting in similar labels. Although, the dataset created will be good for optimization heuristics used by AutoDSE but it may result in erroneous predictions when used as evaluator for heuristics like simulated annealing, multi-armed bandits etc.

The authors are thus suggested to provide a proof-of-concept how the proposed dataset is representative of all “good” and “bad” samples in the entire search space and it will not lead into the issue of out-of-distribution even when the kernel is seen before. I am willing to increase the score if this issue can be mitigated reasonably well during the rebuttal period.

**Relation To Prior Work:**

Reasonably well.

**Summary And Contributions:**

The authors propose to generate a comprehensive set of benchmarks for FPGA targeted high level synthesis which is essentially related to the combinatorial optimization problem of finding an optimal pragma configuration for a given HLS kernel. Naturally, this makes the problem hard and various exploration heuristics have been proposed to date for solving this problem.

From reviewer pov, there are two merits in creation of this dataset: 1) Better evaluation of existing heuristics search and techniques for the given problem (although only benchmarks are needed, no labeling and running the HLS is needed and thus better suited for EDA conferences rather ML-based venues) and 2) Build a regressor out of the labeled dataset to be used as a surrogate/proxy model for evaluating the QoR of a design (pragma configuration of a given kernel) while performing DSE.


The reviewer finds the project to be relevant but needs a good amount of justification on the number of datapoints used which is around ~1000 per kernel out of a huge exponential search space. This is one of the major limitation of this existing work and I will explain in details in “Limitations” section on how this can be a crucial factor in learning a better quality regressor which can be used with any domain-independent heuristics (not just with AutoDSE[1])


1) Atefeh Sohrabizadeh, Cody Hao Yu, Min Gao, and Jason Cong. Autodse: Enabling software programmers 390 to design efficient fpga accelerators. ACM Transactions on Design Automation of Electronic Systems 391 (TODAES), 27(4):1–27, 2022.

---

> ### Author Response · Authors · 2023-08-22
> **Thank you very much for your response!**
>
> We would like to clarify the subsequence embedding generation method: ”The starting token’s embedding is taken as the sub-sequence level embedding.” This is due to the fact that we use a special token indicating “the start of a sub-sequence” and prepend it to the beginning of each subsequence, and thus this token’s embedding represents the entire subsequence after the transformer encoder.
>
> We would like to clarify Section 5.2. We mean 70% designs out of the held-out kernel. For each held-out kernel, we will split it into 70% and 30%: The 30% designs are used for final testing on that held-out kernel; Within the 70% designs, we randomly take 10 designs used for adapting the model. Adaptation means fine-tuning, which is done by fine-tuning both the encoder and the decoder on the randomly chosen 10 designs. Our released source code provides details for doing that (src/adapt.py of the supplementary material).
>
> Regarding your main concern, we would like to provide the following response:
>
> (1) The full design space for the 2mm kernel has more than 492M points. There is no way to run HLS for all the design points (cost for evaluating each design point). This full design space is considered as the entire population.
>
> (2) Any training dataset used in the machine learning algorithm is considered as a sample from the entire population. For sampling strategy, one common strategy is uniform sampling, where each sample has an equal chance to be sampled. This can largely reflect the population distribution, and reduce the OOD issue in the test set. However, this sampling strategy can hardly be implemented in practice (since bias always exists) and has a problem for some applications. For example, positive classes (e.g., cancer) are extremely scarce in healthcare applications. When preparing the benchmark dataset in those applications, the minority classes will be intentionally sampled with a higher probability. Similarly, in our task, the majority of the design points are "bad" designs, and we want to provide more "good" designs with a heuristic-based selection procedure.
>
> (3) This will cause potential OOD for test design points. This requires ML algorithms to overcome the bias in the training dataset and be generalizable to unseen design points. In other words, our dataset can serve as a benchmark dataset to test ML algorithms' OOD prediction power.
>
> (4) That being said, we plan to further improve our dataset by adding more representative samples, to reduce the possibility that certain region of the design space is not covered. We plan to run clustering algorithms for all pragma configurations according to their embedding and select K samples from each cluster.

---

> > ### Comment · Reviewer_mcz1 · 2023-08-22
> > **Thank you for your response**
> >
> > The authors have resolved some of the concerns in their rebuttal. However, one of my concern remains partially addressed related to sampling "representatives" from entire search space. Although, I agree with the authors that the distribution of "good" samples in the search space is skewed and thus heuristics have been applied to over-sample them, bias will be introduced. Since the predictor built on top of this dataset is going to be used as "proxy" for evaluating data-points in black-box optimization, relying entirely on ML algorithms to detect OOD does not solve this issue.
> >
> > I would encourage the authors to go through [1] and devise a methodology for sampling representatives. I have increased the score by +1.
> >
> > [1] Pu, Yewen, et al. "Selecting representative examples for program synthesis." International Conference on Machine Learning. PMLR, 2018.

---

### Official Review · Reviewer_qZxJ · 2023-07-28
**A Benchmark for FPGA C/C++ Performance Prediction**

**Rating:** 6
**Confidence:** 3
**Clarity:** The paper is clear and easy to follow.

**Strengths:**

1. The collected design points are one order of magnitude larger than previous databases. The performance prediction aims to predict both the validity (classification) and quality (5-target regression) of the design.
2. Various SOTA baseline methods (i.e., 10 methods in total) are used for experiments on HLSyn. In addition, detailed evaluations are provided.

**Additional Feedback:**

My feedback is summarized in "Opportunities For Improvement".

**Correctness:**

Yes, the dataset is collected and constructed in a sound way and the evaluation methods and experiment design sounds correct.

**Documentation:**

Yes, the paper provides sufficient details on data collection and experiment setup.

**Ethics:**

No ethical concerns.

**Limitations:**

Yes, this paper discussed its limitations and future work in the last section.

**Opportunities For Improvement:**

1. The motivation of C/C++ performance prediction is to save the time-consuming HLS process. However, there is no metric like run time in this benchmark.
2. The tokens of the design are given without an explanation.
3. Considering that such kinds of databases have been published in the EDA community like DAC and ICCAD, can a conference such as DAC/ICCAD be a more suitable place for publishing this paper?

**Relation To Prior Work:**

Yes, the 2nd paragraph of "Related Work" section clarifies the differences.

**Summary And Contributions:**

This paper presents a benchmark, dubbed HLSyn, for FPGA C/C++ performance prediction consisting of over 42,000 labeled designs covering multiple domains. Experiments across 10 SOTA methods using source code or assembly code graphs show valuable insights.

---

> ### Author Response · Authors · 2023-08-22
> **Thank you very much for your review!**
>
> We would like to clarify the “tokens” for the designs. They are defined for the source code modality, and we obtain the tokens of a design by using a tokenizer on the source code sequence. Specifically, the codet5 model has its tokenizer released by Salesforce whose url link can be found in our paper.

---

### Decision · Program_Chairs · 2023-09-22

**Decision:**

Accept (Poster)

**Comment:**

All the reviewers are overall positive about the paper. ACs don't see obvious reason to overturn the ratings.